**Data Availability Statement:** All relevant data are within the manuscript and its Supporting information files.

# Nitrate and nitrite exposure leads to mild anxiogenic-like behavior and alters brain metabolomic profile in zebrafish

**Manuel García-Jaramillo**[1,2,3,4☯]*, **Laura M. Beaver**[1,2☯], **Lisa Truong**[5], **Elizabeth R. Axton**[2,6¤a], **Rosa M. Keller**[1], **Mary C. Prater**[1¤b], **Kathy R. Magnusson**[2,7], **Robyn L. Tanguay**[5], **Jan F. Stevens**[2,6], **Norman G. Hord**[1,8]

**1** School of Biological and Population Health Sciences, College of Public Health and Human Sciences, Oregon State University, Corvallis, Oregon, United States of America, **2** Linus Pauling Institute, Oregon State University, Corvallis, Oregon, United States of America, **3** Department of Chemistry, Oregon State University, Corvallis, Oregon, United States of America, **4** Helfgott Research Institute, National University of Natural Medicine, Portland, Oregon, United States of America, **5** Department of Environmental and Molecular Toxicology, Sinnhuber Aquatic Research Laboratory, Oregon State University, Corvallis, Oregon, United States of America, **6** Department of Pharmaceutical Sciences, College of Pharmacy, Oregon State University, Corvallis, Oregon, United States of America, **7** Department of Biomedical Sciences, Carlson College of Veterinary Medicine, Oregon State University, Corvallis, Oregon, United States of America, **8** OU Health, Harold Hamm Diabetes Center, Department of Nutritional Sciences, College of Allied Health, University of Oklahoma Health Sciences Center, Oklahoma City, Oklahoma, United States of America

☯ These authors contributed equally to this work.
¤a Current address: The Jackson Laboratory, Sacramento, California, United States of America
¤b Current address: Department of Foods and Nutrition, College of Family and Consumer Sciences, University of Georgia, Athens, Georgia, United States of America
* manuel.g.jaramillo@oregonstate.edu

## Abstract

Dietary nitrate lowers blood pressure and improves athletic performance in humans, yet data supporting observations that it may increase cerebral blood flow and improve cognitive performance are mixed. We tested the hypothesis that nitrate and nitrite treatment would improve indicators of learning and cognitive performance in a zebrafish (*Danio rerio*) model. We utilized targeted and untargeted liquid chromatography-tandem mass spectrometry (LC-MS/MS) analysis to examine the extent to which treatment resulted in changes in nitrate or nitrite concentrations in the brain and altered the brain metabolome. Fish were exposed to sodium nitrate (606.9 mg/L), sodium nitrite (19.5 mg/L), or control water for 2–4 weeks and free swim, startle response, and shuttle box assays were performed. Nitrate and nitrite treatment did not change fish weight, length, predator avoidance, or distance and velocity traveled in an unstressed environment. Nitrate- and nitrite-treated fish initially experienced more negative reinforcement and increased time to decision in the shuttle box assay, which is consistent with a decrease in associative learning or executive function however, over multiple trials, all treatment groups demonstrated behaviors associated with learning. Nitrate and nitrite treatment was associated with mild anxiogenic-like behavior but did not alter epinephrine, norepinephrine or dopamine levels. Targeted metabolomics analysis revealed no significant increase in brain nitrate or nitrite concentrations with treatment. Untargeted metabolomics analysis found 47 metabolites whose abundance was significantly altered in the brain with nitrate and nitrite treatment. Overall, the depletion in brain metabolites is plausibly

**Funding:** This work was supported in part by Celia Strickland and G. Kenneth Austin III Endowment (NGH), the Oregon Agricultural Experimental Station and OSU College of Pharmacy Faculty Development Funds (JFS). It was also supported by National Institutes of Health grants 1S10RR027878-01 (JFS), NCCIH 2R90AT008924-06 (MGJ), and NIEHS Environmental Health Sciences P30 ES030287 (RLT). The content is solely the responsibility of the authors and does not necessarily represent the official views of the National Institutes of Health. The funders had no role in study design, data collection and analysis, decision to publish, or preparation of the manuscript.

**Competing interests:** The authors have declared that no competing interests exist.

associated with the regulation of neuronal activity including statistically significant reductions in the inhibitory neurotransmitter γ-aminobutyric acid (GABA; 18–19%), and its precursor, glutamine (17–22%). Nitrate treatment caused significant depletion in the brain concentration of fatty acids including linoleic acid (LA) by 50% and arachidonic acid (ARA) by 80%; nitrite treatment caused depletion of LA by ~90% and ARA by 60%, change which could alter the function of dopaminergic neurons and affect behavior. Nitrate and nitrite treatment did not adversely affect multiple parameters of zebrafish health. It is plausible that indirect NO-mediated mechanisms may be responsible for the nitrate and nitrite-mediated effects on the brain metabolome and behavior in zebrafish.

## Introduction

Nitrate ($NO_3^-$), a component of leafy green and root vegetables, including beetroot juice (BRJ) and many green leafy vegetables, has blood pressuring-lowering and ergogenic effects in humans [1]. Nitrate supplementation (either as BRJ or sodium nitrate) has also demonstrated benefits pertaining to cardiovascular health [2], such as reducing blood pressure, enhancing blood flow, and elevating the driving pressure of $O_2$ in the microcirculation to areas of hypoxia or exercising tissue [3, 4]. These findings are important to cardiovascular medicine and exercise physiology. Indeed, multiple studies support nitrate supplementation as an effective method to improve exercise performance [5, 6]. Additionally, it has been reported that dietary nitrate can modulate cerebral blood-flow (CBF), decrease reaction time in neuropsychological tests, improve cognitive performance and suggest one possible mechanism by which vegetable consumption may have beneficial effects on brain function in humans [7, 8]. In contrast, other recent studies have found no significant effect of nitrate or nitrite supplementation on cognitive function and this highlights the need for additional studies to clarify the effect of nitrate and nitrite treatment on cognitive function (reviewed in [9, 10]).

Nitric oxide (NO) is a gaseous, free radical signaling molecule produced via enzymatic and non-enzymatic pathways. The enzymatic pathways for NO synthesis are produced by three distinct families of nitric oxide synthase (NOS) enzymes in mammals that use L-arginine and numerous co-factors as substrates [11]. NO conveys essential signaling in the cardiovascular, central nervous, and immune systems [12]. NO, through formation of S-nitrosothiols and nitration of alkenes or other nitrated species, is also considered to have hormone-like properties that take part in different metabolic/endocrine disorders such as diabetes and dysglycemia, thyroid disorders, hypertension, heart failure, and obesity [13]. Furthermore, NO plays an important role in regulation of synaptogenesis and neurotransmission in the central and peripheral nervous system [14, 15]. NO can also be produced by a NO synthase-independent method through the nitrate-nitrite-nitric oxide pathway. Nitrate present in foods or water is reduced endogenously by lingual nitrate reductases in mammals to nitrite ($NO_2^-$) and, in the stomach, to nitric oxide (NO) before distribution via blood to tissues [16, 17]. Several endogenous enzymes, proteins, and chemical species can reduce nitrite to NO including deoxygenated hemoglobin, xanthine oxidoreductase, deoxymyoglobin, mitochondrial enzymes, ascorbic acid, etc. [18]. In spite of the vast amounts of research on NO production, NO-related signaling mechanisms, and the effects of nitrate supplementation on the cardiovascular system; there is still a gap in knowledge regarding whether dietary nitrate supplementation affects the brain metabolome, learning, and other brain functions.

In order to determine the physiological and cognitive effects derived from nitrate and nitrite exposure, we carried out a study with the aquatic model organism *Danio Rerio*

(zebrafish). Zebrafish was chosen because it is a complex vertebrate organism that was originally established as a prime model for developmental studies and, is increasingly used for behavioral neuroscience research in part because of standardized and high throughput behavioral performance assays [19–23]. Importantly, as in humans, the nitrate-nitrite-nitric oxide pathway and NOS enzymes play important roles in regulating NO levels, along with cardiac and blood vessel development in zebrafish [24]. In addition, high genetic homology exists between zebrafish and humans for genes associated with disease [25, 26]. Furthermore, we established that nitrate treatment in zebrafish improves the oxygen cost of exercise [27] as had been observed in humans. While conducting these experiments we also sought to test the hypothesis that nitrate and nitrite treatment would improve indicators of learning and cognitive performance. We also investigated the effects of nitrate and nitrite treatment on zebrafish behavior and the brain metabolome with the aim of elucidating mechanisms that may contribute to the potential improvement of cognitive performance. While these hypotheses are in keeping with nitrate literature in humans, there is also concern that nitrate and nitrite exposure from pollution has adverse effects on fish [28]. To this end, adult zebrafish were exposed to sodium nitrate, sodium nitrite, or control water and tested for changes in learning, memory, and behavior. Furthermore, we utilized targeted and untargeted liquid chromatography-tandem mass spectrometry (LC-MS/MS) analysis to examine the extent to which treatment resulted in changed nitrate or nitrite concentrations in the brain and altered the brain metabolome.

## Materials and methods

### Fish husbandry

Wild type zebrafish (5D) were raised and maintained at the Sinnhuber Aquatic Research Laboratory (SARL) at Oregon State University on standard lab diet (Gemma Micro. Skretting, Tooele, France) in accordance with protocols approved by the Oregon State University Institutional Animal Care and Use Committee (IACUC). Adult fish were maintained at six fish per tank (3 male and 3 female) in 4-liter of aerated water in metal tanks. Experiments were conducted in several cohorts of healthy adult fish, fish from each cohort were all the same age and equally distributed between all treatment groups (cohorts ranged from 9–16 months in age, S1 Fig). Fish water was made with reverse-osmosis water supplemented with Instant Ocean® (Spectrum Brands Blacksburg, VA) at 1.4 g of salt/gallon of water and conductivity between 500–600 μS. Experiments contained three treatment groups which were treated for up to 31 days as 1) no treatment (control fish); 2) sodium nitrate-exposed fish (606.9 mg $NaNO_3$ / L of water); and 3) sodium nitrite-exposed fish (19.5 mg $NaNO_2$ / L of water). The nitrite dose was chosen because it increased blood nitrite levels and improved exercise performance, but was not associated with adverse effects at pathology with the exception of some mild irritation of gill epithelium [27, 29, 30]. For labeling experiments, a subset of fish was switched to water containing >99% stable isotopes of $Na^{15}NO_3$, or 100% $Na^{15}NO_2$ (Cambridge Isotope Laboratories, Tewksbury, MA) at day 28 for 3 days of treatment prior to collection. Nitrate and nitrite were dissolved in freshly prepared fish water and, unless otherwise indicated, chemicals were purchased from Sigma-Aldrich (St. Louis, MO). The fish water and treatment exposure were replaced every 36 hours throughout the duration of the experiment to maintain low ammonia levels and consistent treatments; pH was held at 6.8–7, total ammonia levels to 0–2.0 ppm, and temperature at 27–29°C. Fish were fed a standard lab diet twice a day (Gemma Micro. Skretting, Westbrook, ME) at a volume of ~3% body weight/day. All efforts were made to ameliorate potential pain and distress including post-behavior assessments, tested zebrafish are housed together, and monitored for 24 hours for morphological

(e.g. adhesions, bruising or abrasion to the skin) and abnormal swim defects. If any zebrafish exhibit these effects, they are kept in isolation and observed according to OSU Animal Care and Use Protocol. For sample collections fish were primarily euthanized with an overdose of the anesthesia drug tricaine mesylate, and all efforts were made to minimize suffering. Fish were then dried, weighed, measured for standard length, and brains were collected and snap frozen in liquid nitrogen. Fished used in hormone analysis were euthanized with rapid cooling, followed by the secondary method of cervical dislocation, and frozen in liquid nitrogen [31]. Samples were stored in -80°C until used for analysis.

## Nitrate and nitrite quantification in water

Water was collected during the first week of the experiment and saved directly after a water change (designated as fresh), or 36 h post water change (designated as used). For nitrate measurements, fish water was snap frozen directly. For nitrite measurements, 1 mL fish water was mixed with 250 μL of a stop solution (containing potassium ferricyanide, N-ethylmaleimide, NP-40) as previously published [32]. Nitrate and nitrite concentrations were determined by ozone chemiluminescence as previously described on a Sievers Nitric Oxide Analyzer (NOA; Zysense, Frederick, CO) [30, 33].

## Behavioral assays

Each zebrafish underwent behavioral assessments through a circuit of 1) swim behavior, 2) startle response and 3) learning and memory assays, where for each run, 4 fish per treatment were included. Multiple runs occurred over 4 days, between 14–17 days of treatment, and fish were within 4 h of being feed when assays were conducted. Swimming behavior and startle response was tested using a zebrafish visual imaging system (zVIS) as previously described in individual fish [34, 35]. Briefly, in the free swim assay fish were placed in a tank with 1.7L of water and the data from the first minute was ignored. The location of the fish was then analyzed by region of tank (top, middle, bottom) for the following 7 minutes (stressed, novel tank environment during minutes 1–8), and then during the last 7 minutes of the assay (minutes 11–18) speed and distance fish traveled was measured (unstressed environment). Habituation to an audio startle stimulus was tested in an array of 8 tanks (12cm × 12cm) filled with 750 mL of fish water [34]. Taps were generated by an electric solenoid below each tank. Following a 10-minute acclimation period, a total of five taps were delivered, with 20s following each tap, and the distance moved between taps was quantified.

Custom-built shuttle boxes were used to test learning with a modified protocol as previously described [34, 36]. The programmed protocol of this active avoidance conditioning test was designed to condition the zebrafish to leave the compartment with blue light ("reject side") and swim to the dark side ("accept side", also referred to as the correct side). The source of light is not at a fixed location in the box but rather depending on where the fish is in the box, the light opposite of the fish is turned on. There were a total of 30 trials; each trial consisted of giving the zebrafish 8 seconds to "seek" a dark side of the tank after the blue light came on to avoid a moderate shock. If the fish did not move to the correct side, the 16 second (s) shock period was initiated. A moderate pulse of 5 V was delivered at 1 s intervals, for a duration of 500 ms. Fish were removed from the assay when they did not swim to the correct side during 8 consecutive trials and these fish were counted as repeatedly failed. The statistical method remained as previously described [36], with the data fit using linear regression models to calculate the initial performance of the fish (intercept) and the rate of learning (slopes) for each recorded parameter including the period of time to decision and time shocked [36].

## Epinephrine, norepinephrine, and dopamine quantification

Stress hormones epinephrine, norepinephrine, and their precursor dopamine were measured in fish (n = 12) using the 3-CAT ELISA (Rocky Mountain Diagnostics, Inc., Colorado Springs, CO) per manufacturer's recommendations. Snap frozen whole zebrafish were ground in liquid nitrogen with mortar and pestle. To normalize variations in fish weight, the resulting whole fish powder was mixed with a buffer at a ratio of 100 mg fish powder to 500 μL HCL buffer solution, containing EDTA and sodium metabisulfite. Samples were centrifuged for 20 minutes at 10,000 × g at 4°C and supernatants collected. A standard curve was generated for each compound concentrations of 0.5, 1.5, 5, 20, and 80 ng/mL, for epinephrine and dopamine, and 0.2, 0.6, 2.0, 8.0, and 32.0 ng/mL for norepinephrine. Samples were diluted 1:1 to be in the range of the standard curve. A Spectramax® M2 plate reader (Molecular Devices, Sunnyvale, CA) was used to measure concentration at 450 nm.

## Extraction of zebrafish brains for analysis

Twelve brains per treatment group were snap frozen using liquid nitrogen after four weeks of treatment and two brains were pooled together to compose each sample. Each sample was added into 2 mL pre-filled tubes containing 300 mg of RNAse and DNAse free zirconium oxide beads (0.5 mm diameter, ceria stabilized, Next Advance, Averill Park, NY). A mixture of 80:20 methanol: water at -80°C was used as the extraction solvent as previously described [37]. Brains were homogenized with a bullet blender (Precellys® 24-bead-based homogenizer for 2 minutes at 1350 rpm). Extracts were incubated at -20°C for 1 hour and then centrifuged at 13,000 rpm (Eppendorf, Hauppauge, NY) and 4°C for 10 min. The supernatant was split into three 1.5 mL Eppendorf tubes: 100 μL was aliquoted for nitrate and nitrite isotope targeted analysis by LC-MS/MS; 200 μL was aliquoted for untargeted metabolomics analysis, and the remainder (variable volume) was reserved and stored at -80°C.

## Targeted LC-MS/MS analysis of nitrate and nitrite

In order to quantify nitrate and nitrite uptake into the brain, we used a previously described LC-MS/MS approach [38]. Assessing the percent enrichment ($^{15}N/(^{15}N+^{14}N)$ x 100%) allows us to determine the proportion of the nitrate and nitrite that was derived from exogenous sources (stable isotope treatment in water) versus endogenous source (nitrate oxidized from NO produced by NOS enzymes). This method utilizes 2,3-diaminonaphthalene (DAN) derivatization, which reacts with nitrite under acidic conditions to produce 2,3-naphthotriazole (NAT). The production NAT was measured with the previously described method [38] with minor modifications. Briefly, NAT was chromatographically separated on an InfinityLab Poroshell 120 HPH-C18 column (2.7 μm, 2.1 × 50 mm, Agilent, Santa Clara, CA), in a run time of 10 minutes, and detected using a multiple reaction monitoring (MRM) method on an ABSciex 3200 QTRAP mass spectrometer operated in positive ionization mode. Mass spectrometry allows for the quantification of $^{14}N$-NAT (*m/z* 170.1) and $^{15}N$-NAT (*m/z* 171.1). The percent enrichment (%) was calculated as: $[^{15}N/(^{15}N+^{14}N) \times 100]$.

## Untargeted LC-MS/MS metabolomics analysis

Aliquoted extracts were sonicated for 5 minutes and clarified by centrifugation at 13,000 rpm for 10 minutes. The supernatant was transferred to glass mass spectrometry vials and LC-MS/MS-based metabolomics was performed as previously described [27, 39]. Briefly, ultra-high-pressure liquid chromatography (UPLC) was performed on a Shimadzu Nexera™ system (Shimadzu, Columbia, MD) coupled to a quadrupole time-of-flight mass spectrometer (AB SCIEX

TripleTOF 5600). Chromatographic separations were conducted on an Inertsil® Phenyl-3 column (4.6 × 150 mm, GL Sciences, Torrance, CA). Elution was achieved using a binary gradient employing as solvent A water, and solvent B methanol, both containing 0.1% formic acid (v/v), as described previously [39]. LC-MS/MS conditions were adapted from Kirkwood et al. (2012) [39] with some modifications. The gradient started with 5% B and was held for 1 min at 5% B, followed by a 11-min linear gradient from 5% to 30% B. The gradient was increased linearly to 100% B at 23 min, held for 5 min at 100% B and, finally, stepped back to 5% B to equilibrate the column. The flow rate was 0.4 mL/min. The auto-sampler temperature was held at 10˚C, the column oven temperature at 50˚C, and the injection volume was 5 μL. Time-of-flight (TOF) mass spectrometry (MS) was operated with an acquisition time of 0.25 s and a scan range of 70–1200 Da. Tandem mass spectrometry (MS/MS) acquisition was performed with collision energy set at 35 V and collision energy spread of 15 V. Each MS/MS scan had an accumulation time of 0.17 s and a range of 50–1250 Da using information-dependent MS/MS acquisition (IDA). Ion source gas 1 and 2 and curtain gas (all nitrogen) were set at 50, 40, and 25, respectively. The source temperature was set at 500˚C and the ion spray voltage at 4.5 kV in positive ion mode. The mass calibration was automatically performed every 6 injections using an APCI positive/negative calibration solution (AB SCIEX) via a calibration delivery system (CDS). A separate quality control (QC) pool sample was prepared by combining 5 μL of each sample. Quality control was assured by: (i) randomization of the sequence, (ii) injection of QC pool samples at the beginning and the end of the sequence and between each 10 actual samples, (iii) procedure blank analysis.

## Untargeted metabolomics data processing

Raw data was imported into PeakView™ with XIC Manager 1.2.0 (ABSciex, Framingham, MA) for peak picking, retention time correction, and peak alignment. Metabolite identities were assigned as previously described by matching with an in-house library consisting of IROA standards (IROA Technology, Bolton, MA) and other commercially available standards (650 total) [27]. The peak list was exported to MultiQuant 3.0.2 to integrate chromatograms to obtain peak area values for all of the assigned metabolites.

## Statistical analysis

To determine significant differences between three treatment group data were analyzed using a one-way ANOVA with Tukey post hoc test (*P*-value < 0.05, statistically significant) with GraphPad Prism 4 software (La Jolla, CA). Significant differences were calculated with two-way ANOVA and Tukey post hoc test or a repeated measures two-way ANOVA and Tukey post-hoc test (*P*–value < 0.05, statistically significant) when both treatment and another condition, like water condition, zone of tank, or a behavioral stimulus, was present [40]. For the shuttle box assay a linear regression was fitted to the data for each treatment to generate initial time and rate of learning graphs, while a separate analysis of variance (AOV) followed by a Tukey's statistical difference was used to calculate statistical significance amongst the groups [36]. For metabolomics data, annotated metabolites were used to conduct multivariate statistical analysis. Pathway analysis and partial least squares-discriminant analysis (PLS-DA), were generated with MetaboAnalyst 4.0 [41]. The significance of individual metabolites between the treatment groups was assessed with a one-way ANOVA followed by Fisher's post-hoc analysis and Holm FDR-correction, with a *P*-value of < 0.05 and a *q*-value <0.1 indicating significance. If needed, data were logarithmically transformed to correct for unequal variance or non-normal distribution. No outliers were excluded from the statistical analyses. Figures were

generated with Prism 8 (GraphPad Software, San Diego, CA), PowerPoint 2016 (Microsoft, Redmond, WA), and MetaboAnalyst 4.0 [41].

## Results

### Effect of nitrate and nitrite treatment on health parameters and learning

Treatment increased nitrate or nitrite levels in the fresh and used fish water (Fig 1A and 1B). Furthermore, both nitrate and nitrite concentrations in control water were maintained at low levels throughout the treatment period (Fig 1A and 1B). Several parameters of fish health, including fish length and weight, were not significantly changed with nitrate or nitrite treatment (Fig 1C and 1D). Likewise, no significant differences were found between treatment groups for the distance and velocity fish traveled in an unstressed environment (Fig 1E and 1F, $P = 0.2089$ and 0.2088, respectively). No differences were detected in habituation to a startle with nitrate or nitrite treatment, but nitrate-treated fish had a slight reduction (10%) in startle response as measured by significantly less distance traveled (Fig 1G).

In order to address if nitrate and nitrite treatments altered learning, fish were tested in a learning and memory assay using custom-built shuttle box, where over 30 consecutive trials they learned to avoid an adverse event (mild shock) by moving when a light came on (Fig 2A). As seen from the linear regression calculated from the data, both nitrate and nitrite treated fish initially took longer to make a decision and were shocked longer (Fig 2B). Over subsequent trials, more nitrate- and nitrite-treated fish (5–7% of the population tested) had to be removed from the assay because they repeatedly failed to learn (Fig 2C). However, as the experiment was repeated over multiple trials both nitrate- and nitrite-treated fish were able to learn as seen in improvements in decision time, time shocked, and rate of learning over trials (Fig 2D). These observations from the linear regression hold when analyzing the data for statistical significance; overall, nitrate and nitrite treatment significantly increased the percentage of fish that failed to make a decision and were shocked (Fig 2E). When the population of fish analyzed was filtered to include only fish that could learn (i.e., completed the assay), the nitrite-treated fish were no longer significantly impaired but significant deficits were still present in nitrate-treated fish for time shocked and time to decision (Fig 2E).

### Effect of nitrate and nitrite treatment on behavior and catecholamine levels

The effect of nitrate and nitrite exposure on behavior was tested in the free swim assay where fish were placed in a novel tank. As expected, control fish spent similar amounts of time at all three depths of the tank, balancing safety from predation and opportunity to find food (Fig 3A). In contrast, there was a significant difference between the time the nitrate- and nitrite-treated fish spent between the bottom and top zones (Fig 3A). Thus nitrate- and nitrite-treated fish spent 22–35% more time in the bottom zone, as compared to control fish, although there was not a significant treatment effect when just the bottom zone was considered. Nevertheless, the increase in bottom-dwelling is suggestive of mild anxiogenic-like behavior. Since anxiety can be associated with stress, we measured the levels of some stress hormones and found neither nitrate, nor nitrite treatment significantly increased epinephrine or norepinephrine levels (Fig 3B–3C). It appeared that nitrite-treated fish experienced lower concentrations of these hormones yet high variability between fish led to no significant differences being detected. Nitrate or nitrite treatment also did not significantly change dopamine concentrations which is the precursor for epinephrine and norepinephrine (Fig 3D).

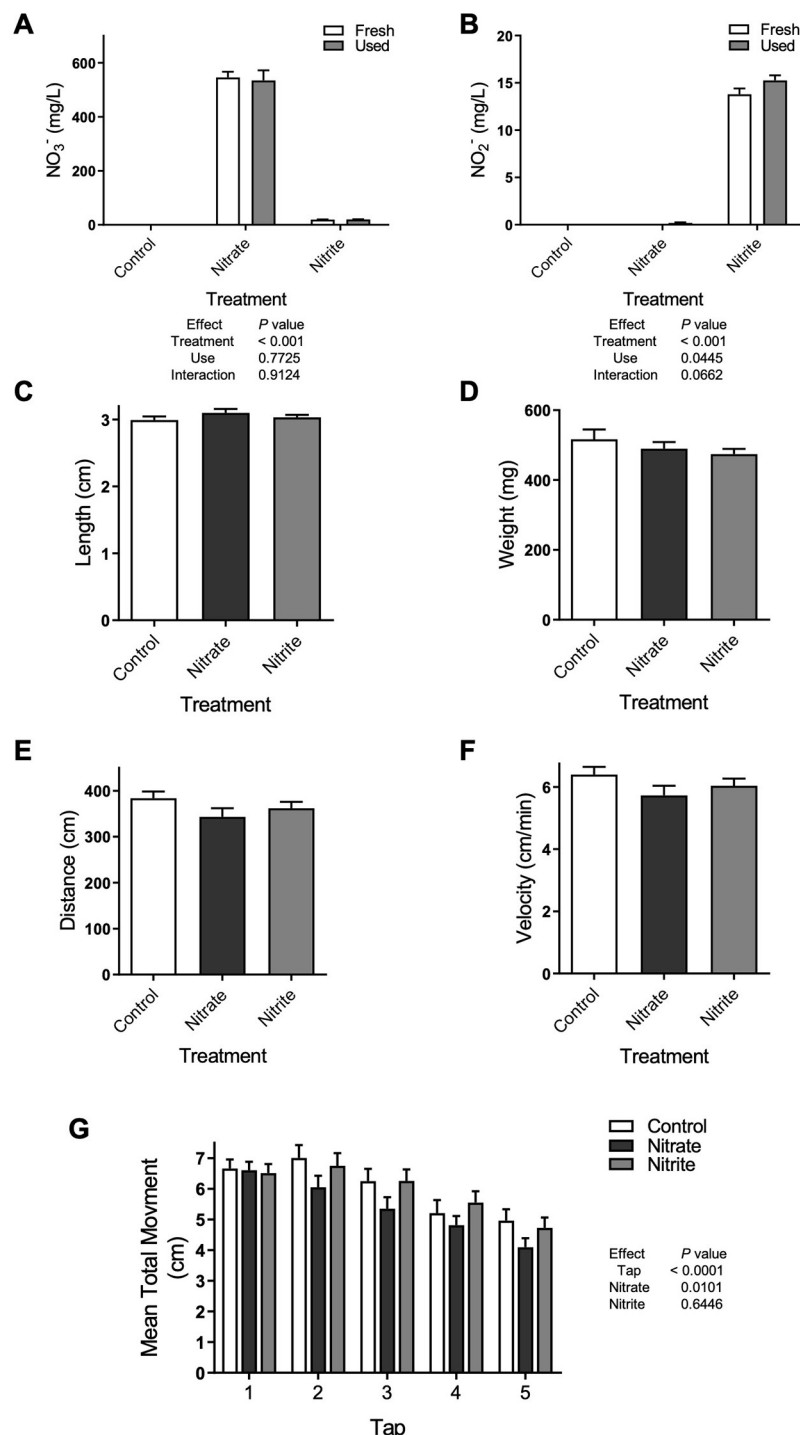

**Fig 1. Nitrate and nitrite treatment did not adversely affect multiple parameters of health but nitrate treatment significantly decreased movement following a startle.** Adult zebrafish were treated with control water, sodium nitrate, or sodium nitrite and (A) nitrate and (B) nitrite concentrations were measured in newly treated fish water (fresh) and water at the end of 42-hour use (used) (n = 7–10). Fish (C) length and (D) weight was measured after 28–31 days of treatment (n = 18–33). At 14–17 days of treatment the (E) distance and (F) velocity zebrafish traveled was quantified in the voluntary swimming assay (n = 39–42) or (G) the response to five sequential acoustic startles (as taps against the fish tank) was quantified (n = 84–90). (A-G) Bars represent the mean ± SEM.

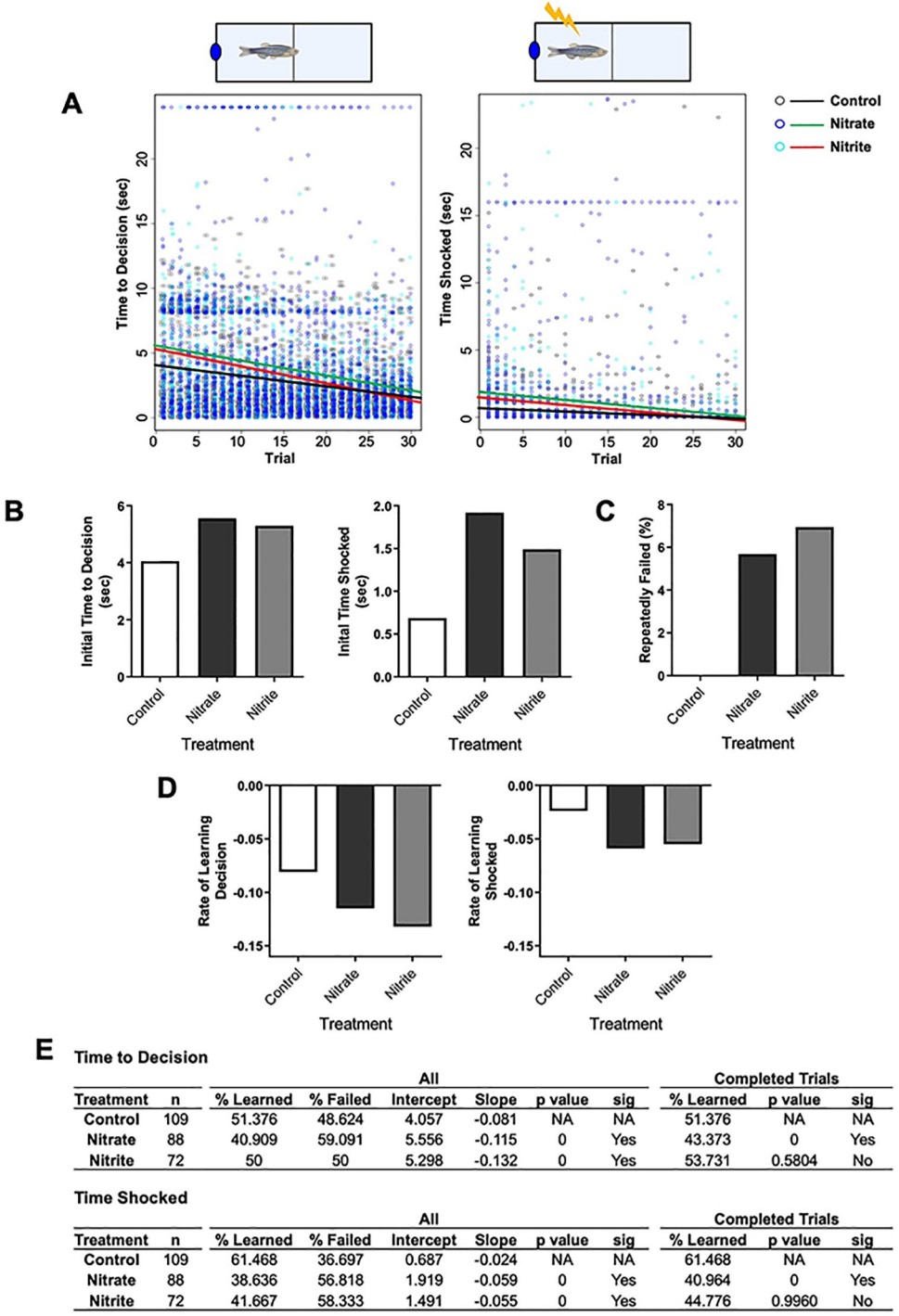

**Fig 2. Nitrate and nitrite treatment were associated with an initial decline in learning, but fish learned over repeated tests.** Adult zebrafish were treated with control water, sodium nitrate, or sodium nitrite for 14–17 days when learning and memory were tested in the shuttle box assay (n = 72–109). (A) Time-to-decision and time shocked was recorded for each fish and trial (dots) and linear regression of the data were calculated (lines). As calculated from the linear regression the bars indicate (B) initial periods of time fish spent for the indicated measure and (D) rates of learning as quantified by the slope from the linear regression. (C) Bars indicate the percentage of fish that were removed from the assay because they did not swim to the correct side during eight consecutive trials. (E) Statistical summary of shuttle box results as calculated by an analysis of variance (AOV) followed by a Tukey's post-test where "All" indicates data from all fish analyzed, while "Completed trials" excludes data from fish that repeatedly failed.

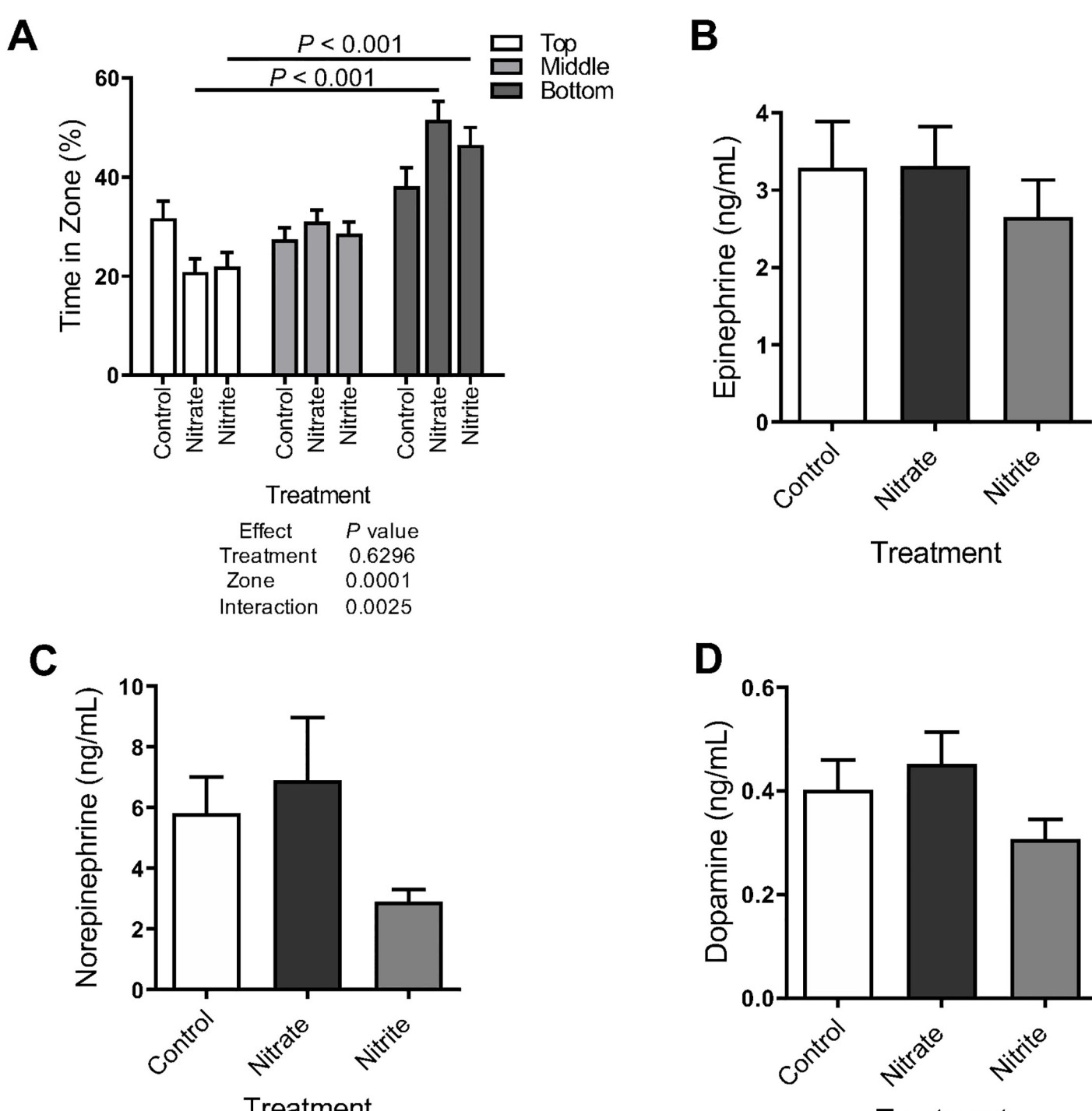

**Fig 3. Nitrate and nitrite treatment increased mild anxiogenic-like behavior in zebrafish.** Adult zebrafish were treated with control water, sodium nitrate, or sodium nitrite for 14–17 days. (A) Movement in a novel tank was recorded and the percent of time spent in the bottom, middle and top zones of tank are indicated (n = 83–89). (B-D) Concentrations of hormones were measured in whole fish by ELISA (n = 12). (A-D) Bars represent the mean ± SEM.

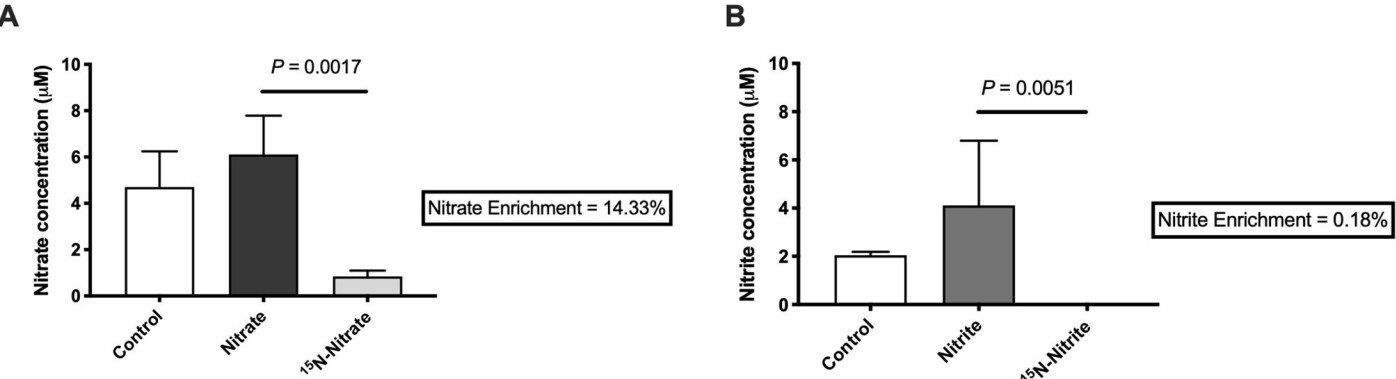

**Fig 4. Little uptake of nitrate or nitrite from treatments was found in the brain.** The concentration of (A) nitrate or (B) nitrite was measured using targeted LC-MS/MS in control animals or animals treated with (A) unlabeled and $^{15}$N-labeled nitrate, or (B) unlabeled and $^{15}$N-labeled nitrite. (A-B) Zebrafish brains were collected on day 31 and percent enrichment (in boxes), indicates the relative amount of nitrate or nitrite in the brain that was derived from the treatment. Bars represent the mean concentration ± SEM (n = 6).

### Nitrate and nitrite uptake into the brain

For the last three days of the experiment, a subset of fish was treated with $^{15}$N-nitrate or $^{15}$N-nitrite in order to study the uptake of nitrate and nitrite into brain tissue. The resulting percent enrichment results show the proportion of nitrate and nitrite derived from exogenous sources (the treatment in water) versus endogenous sources such as oxidation of NO from NOS-mediated production. We observed a low uptake of nitrate (14%) and almost no uptake of nitrite (0.1%) in the brain, which can be seen by comparing the fish that received labeled nitrate or nitrite as compared to the respective unlabeled nitrate or nitrite treatment conditions (Fig 4A and 4B). Furthermore, no significant changes in nitrate or nitrite concentrations were detected in the brain of animals treated with nitrate or nitrite (Fig 4A and 4B) when compared with the control group. Taken together, these results suggest that the behavioral changes observed with nitrate and nitrite exposure are likely due to indirect effects of treatment on brain metabolism, rather than a direct effect via influx of the nitrate or nitrite into the brain.

### Metabolomics results

One hundred twenty-four (124) metabolites were annotated using our in-house library (S1 Table). Of these metabolites, 47 were significantly changed among at least one treatment group, as compared to the others and FDR-corrected *P*-values (*q*-values) for all significantly changed metabolites, between all treatment groups, are listed in S2 Table. For example, deoxyadenosine diphosphate (dADP) was significantly up-regulated (*q* = 0.018) in fish exposed to nitrate and nitrite, and desmosterol, the immediate precursor of cholesterol in the Bloch pathway of cholesterol biosynthesis, was significantly down-regulated (*q* = 0.018) in fish exposed to nitrate and nitrite.

Partial least squares discriminant analysis (PLS-DA) demonstrates spatial clustering and separation between treatment groups when considering all the annotated compounds (Fig 5A). There are two importance measures in PLS-DA: one is variable importance in projection (VIP) and the other is the weighted sum of absolute regression coefficients. The VIP graph of the most relevant 30 features (when considering the three treatments) is shown in Fig 5B. The colored boxes on the right indicate the relative concentrations of the corresponding metabolite in each group under study. Among the positively correlated metabolites with the

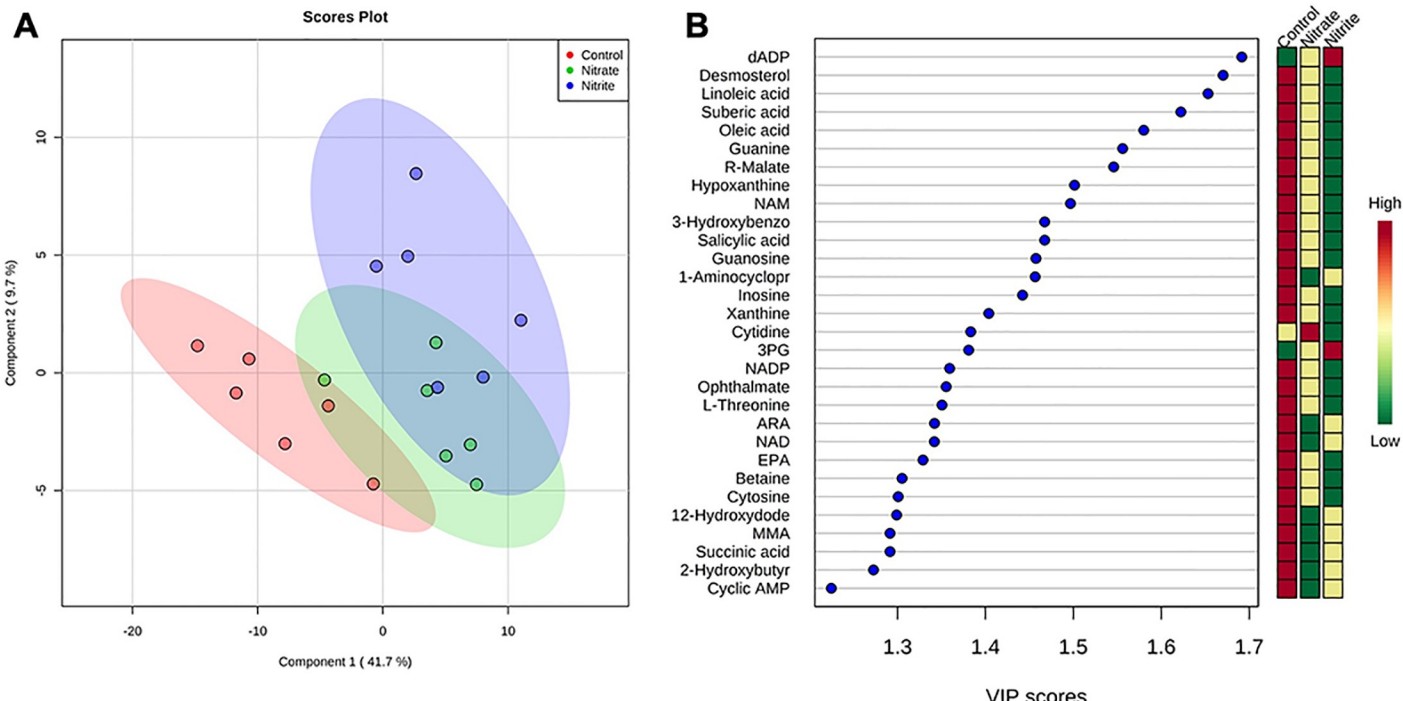

**Fig 5. Nitrate and nitrite treatment significantly altered the abundance of some brain metabolites.** Adult zebrafish were treated with control water, sodium nitrate, or sodium nitrite for 31 days and brain metabolites were measured using untargeted LC-MS/MS. (A) Partial least squares discriminant analysis (PLS-DA) scored plot demonstrates spatial clustering and separation between treatment groups when considering all the annotated compounds. (B) PLS-DA variable importance in projection (VIP) graph of the most relevant 30 features (when considering the three treatments). Colored boxes at right indicate the mean relative concentrations of the corresponding metabolite in each treatment group under study. Red color indicates higher abundance, while green color indicates lower abundance. The PLS-DA model display 95% confidence region. Abbreviations: dADP (deoxyadenosine diphosphate), NAM (N-acetyl-L-methionine), 3-Hydroxybenzo (3-Hydroxybenzoic acid), 1-Aminocyclopr (1-Aminocyclopropane-1-carboxylate), 3PG (3-Phosphoglyceric acid), NADP (Nicotinamide adenine dinucleotide phosphate), ARA (Arachidonic acid), NAD (Nicotinamide adenine dinucleotide), EPA (Eicosapentanoic acid), 12-Hydroxydode (12-Hydroxydodecanoic acid), MMA (Methylmalonic acid), 2-Hydroxybutyrate (2-Hydroxybutyric acid).

highest VIP scores associated with the PLS-DA were dADP, desmosterol, linoleic acid, suberic acid, oleic acid and guanine.

Notably, nitrate or nitrite treatment resulted in significant differences among multiple metabolites involved in purine metabolism like hypoxanthine, xanthine, inosine, guanine, guanosine, deoxyadenosine diphosphate (dADP) and cyclic adenosine monophosphate (cAMP). Interestingly, we also observed a significant decline in nicotinamide adenine dinucleotide phosphate (NADP) and NAD$^+$. We observed a significant depletion in the annotated fatty acids (linoleic acid (LA), eicosapentaenoic acid (EPA), and arachidonic acid (ARA)) with nitrate and nitrite treatments. Remarkably, LA was depleted by 50% by nitrate treatment and by ~90% in the nitrite treatment when considering normalized peak intensity values. Similarly, ARA was depleted by 80 and 60% in the nitrate and nitrite treatments, respectively. We also observed lower abundances for some of the annotated TCA (tricarboxylic acid cycle) cycle intermediates (i.e., malate and succinic acid). A depletion in amino acids threonine, N-acetyl-L-methionine (NAM), and phosphoserine, was observed with nitrate and nitrite treatment. Notably, we observed that nitrate and nitrite treatment had an effect on γ-aminobutyric acid (GABA, Fig 6A), the chief inhibitory neurotransmitter in the developmentally mature central nervous system and its precursor, glutamine (Fig 6B) [42]. Nitrate treatment caused a

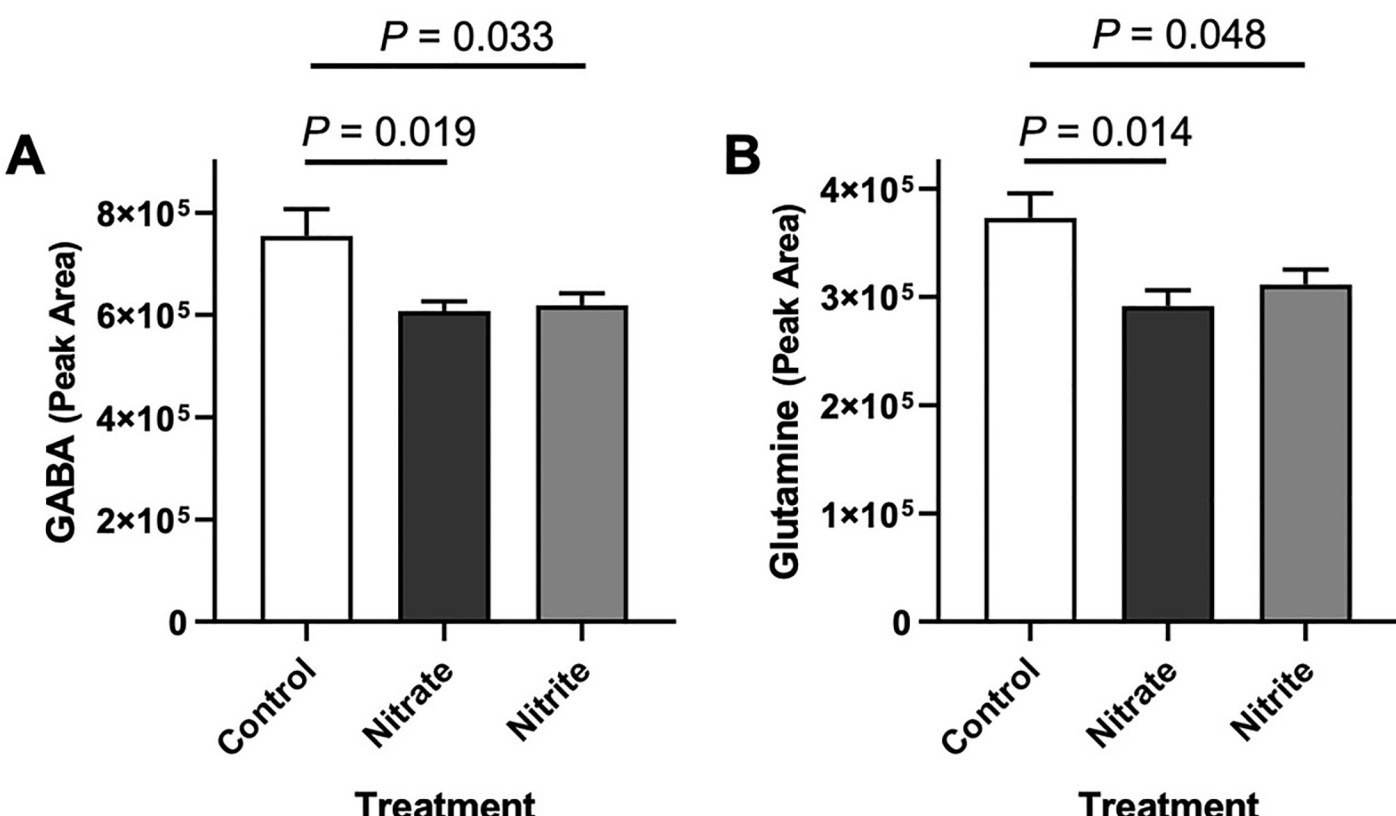

**Fig 6. Nitrate and nitrite treatment decrease gamma-aminobutyric acid (GABA) and glutamine levels in zebrafish brain.** Adult zebrafish were treated with control water, sodium nitrate, or sodium nitrite for 31 days. Abundance of (A) GABA and (B) glutamine was measured in brain tissue by LC-MS/MS. Bars represent the mean peak area ± SEM (n = 6).

significant 22% reduction in the abundance of glutamine and 19% reduction in GABA in zebrafish brains. Nitrite treatment also caused a significant 17% reduction in the abundance of glutamine and 18% reduction of GABA in zebrafish brains. Interestingly, no significant differences in the abundance of the excitatory neurotransmitter glutamate were found with nitrate or nitrite treatments, thus GABA abundance changed in parallel with glutamine, but not glutamate levels.

## Discussion

We did not find evidence to support the hypothesis that nitrate and nitrite treatment would improve indicators of learning and cognitive performance in healthy zebrafish. While nitrate and nitrite treatment did not adversely affect multiple parameters of health, these treatments were associated with mild anxiogenic-like behavior and an initial deficit in learning, which was consistent with either decreased executive function or associative learning. The zebrafish in this study exhibited behavior we interpreted as mild anxiety, as evidenced by staying near the bottom of the novel tank. The mild anxiogenic-like behavior observed in nitrate- or nitrite-treated zebrafish was similar in scale to zebrafish that were not allowed to exercise [43].

Nitrate, but not nitrite treatment, caused a slight reduction in startle reactivity, but no difference was detected in habituation with treatment. We also observed an initial delay in

zebrafish decision making following a light stimulus and increased time being shocked initially in the shuttle box task which could represent an initial deficit in associative learning and/or executive function (e.g., decision making). This is inconsistent with literature that showed nitrate, given as BRJ supplement, improved reaction time and cognitive performance [7, 44–46]. A plausible mechanism underlying nitrate-induced cognitive improvements is increased NO-mediated vasodilation, yielding improved CBF [7, 9, 47, 48]. This is exemplified by a study in older adults where two days of consuming a high nitrate diet increased regional cerebral perfusion in frontal lobe white matter, particularly between the dorsolateral prefrontal cortex and anterior cingulate cortex [47]. These brain regions participate in executive function, which may have been affected by nitrate and nitrite treatment in our study. In contrast with our results, multiple studies show no significant association with foods containing nitrate and changes in cognitive function or mood [49–55]. Possible factors contributing to conflicting cognitive responses reports and our own study include different routes of treatment (continual in water vs episodic in meals), different nitrate doses and lengths of treatment, food matrix effects, age and health status of participants, or unidentified species-specific effects. A significant limitation of this study is the use of whole zebrafish brains to derive metabolomics data, limiting our ability to draw inferences to specific functional structures within the brain, like the zebrafish equivalent of the prefrontal cortex [56].

While we have previously shown the nitrate and nitrite doses used here increased blood and whole-body nitrate and nitrite levels, the treatments were not associated with a significant increase in the concentration of nitrate or nitrate in the brain, and only a minor, or almost no uptake of these chemicals into the brain. Therefore, we propose that the observed changes in behavior due to nitrate and nitrite exposure are due to indirect effects of NO on cell signaling mechanisms in neurons, astrocytes or other cell types due to the observed changes in annotated metabolites. Namely, enhanced signaling through NO-dependent mechanisms, such as enhanced cGMP and NO-dependent S-nitrosylation are associated with changes in brain ion channel function [57]. Our data show that nitrate or nitrite treatment led to modulation of several important brain metabolites including neurotransmitters and their precursors, fatty acids, purine nucleotides and nucleosides, and amino acids, fatty acids. While the evidentiary basis is circumstantial, it is tempting to associate modulation of concentrations of these metabolites with changes in neuronal structure, function and signaling with the observed behavioral changes due to nitrate and nitrite.

The observed reduction of brain GABA and glutamine levels with nitrate and nitrite treatment is noteworthy because perturbations in the GABAergic system have been associated with anxiety and depression and thus may be central to the behavioral changes we observed [58–60]. Glutamine is a substrate for GABA production and serves as an important energy source for the nervous system. We also observed changes in brain purine-related metabolites, which is consistent with the known relationship between exogenous and endogenous pathways that generate NO [18, 24, 27]. Notably, the nitrate- and nitrite-dependent decreases in nicotinamide adenine dinucleotide phosphate (NADP) and $NAD^+$ to recapitulate changes observed in mitochondrial dysfunction and aging in the brain [61]. Physiologically, $NAD^+$ depletion is also observed in response to excessive DNA damage due to free radical attack, resulting in significant poly(ADP-ribose) polymerase (PARP) activation and a high turnover and subsequent depletion of $NAD^+$, as well as in chronic immune activation and inflammatory cytokine production resulting in accelerated CD38 activity and decline in $NAD^+$ concentrations. If nitrate and nitrite exposure were mechanistically linked to decreased $NAD^+$ levels, this could link oxidative damage to low $NAD^+$ concentrations to inflammation and neurological dysfunction in the brain.

The decreased brain concentrations of fatty acids including LA, EPA, ARA observed with nitrate and nitrite treatment are similar to our previous observations in nitrite-treated whole fish [27]. We surmised that nitrite exposure in whole zebrafish led to the stimulation of fatty acid oxidation for energy production. It is unclear why nitrate, and especially nitrite, would selectively stimulate depletion of these fatty acids in brain. These diet-derived fatty acids are a major component of neuronal membranes which are important in brain development and function [62]. Low levels of very long chain PUFAs in the brain also affect the brain dopamine systems and influence risk, along with other genetic factors, of neurological disorders, including Parkinson's disease, schizophrenia and attention-deficit hyperactivity disorder (ADHD). Altering membrane PUFA composition *in vitro* has been shown to alter the function and/or signaling of a variety of receptors, including dopaminergic, cholinergic, and GABAergic receptors, as well as the $Na^+/K^+$ ATPase. Taken together with the observation that nitrate and nitrite exposure decreased the concentration of brain desmosterol, the precursor of cholesterol biosynthesis, it is biologically plausible that cell membrane composition of cholesterol, LA, EPA and ARA could lead to changes in neuron function. These structural changes in membrane composition in neurons could potentially interact with other observed changes in metabolites that could influence behavior, including altered concentrations of GABA and glutamate.

Besides the indirect mechanisms by which nitrate and nitrite could mediate the effects observed in these studies, NO exerts direct neurological effects including acting as an anterograde neurotransmitter, a retrograde neurotransmitter, regulator of presynaptic plasticity in GABAnergic and glutamatergic neurons, and effecting dendritic spine growth (reviewed in [63]). NO is directly involved in learning and memory, and NO modulators are being explored for the treatment of anxiety [64]. Manipulation of brain NO levels in rodents decreased anxiety when specific doses of L-arginine (NO precursor), L-NAME (NOS inhibitor), or sodium nitroprusside (NO donor) were given, but a high dose of L-NAME decreased locomotor activity, similar to our result [65, 66]. Consistent with the observed mild anxiogenic-like behavior, studies in mice showed anxiogenic effects of sildenafil (NO), or the combined treatment of sildenafil and ascorbic acid [67–69]. As manipulation of NO levels in the brain can yield both anxiolytic or anxiogenic effects, it is possible that the changes in behavior we observed may be attributable to high NO levels, but we have not assessed surrogate markers of this phenomenon, such as nitrated tyrosine levels in brain tissue [64].

There is a substantial body of evidence linking intake of diets high in nitrate, as well as pharmaceutical exposure to NO donor drugs, to risk of headache. As such, we could speculate that NO-dependent mechanisms could contribute to the observed changes in zebrafish behavior due to nitrate and nitrite treatment. Organic nitrates, which serve as vasodilatory NO prodrugs, such as nitroglycerin, and sildenafil, a PDE inhibitor that inhibits NO degradation, are each associated with side effects that include headaches. In humans, animal models and fish, organic nitrates like nitroglycerin have been used to model migraines [70, 71]. Organic nitrates cause NO-mediated vasodilation of blood vessels in the brain, and "immediate" mild-to-medium severity headaches or "delayed" migraines are associated with increased cGMP or NO-dependent S-nitrosylation-mediated changes. Headache is also the most common side effect in patients taking sildenafil, which promotes blood flow to organs like the brain, through cGMP. Furthermore, consumption of high nitrite foods was associated with headaches in some people [72]. Migraines have also been correlated in humans with oral microbiomes that increased abundances of nitrate, nitrite, and NO reductase genes supporting that nitrite and NO could promote migraines [73]. Taken together, it is plausible to speculate that nitrate- or nitrite-induced production of NO in blood vessels stimulated vasodilation and caused a headache or migraine.

There are well-established adverse effects of nitrate pollution in aquatic ecosystems, referred to as eutrophication, that affect a variety of species (reviewed in [28]). Likewise, human consumption of nitrate- and nitrite-contaminated water or excessive intake from vegetables may also cause adverse effects [74]. An endocrine-disrupting role of nitrate and nitrite has been observed in various species, and the possible pathways of altering steroidogenesis have been proposed [75]. Both glutamate and GABA are involved in pituitary hormone release in fish. There is also good evidence for the involvement of GABA in luteinizing hormone release in fish [76]. Other studies have indicated that high nitrate and nitrite exposure from drinking water and diet may exert adverse effects on the development of the human nervous system [77, 78]. Nitrate and nitrite can also perturb the activity of dopaminergic (DA) neurons by acting through estrogen receptor (ER) in early development of zebrafish [79] at concentrations around the safety limit for drinking water recommended by the Environmental Protection Agency (EPA) and the World Health Organization (WHO) (10 mg/L $NO_3$-N and 1 mg/L $NO_2$-N, respectively) [80]. While many of these studies were conducted during embryonic development, and are different from own limited adult exposure, they highlight that nitrate and nitrite can have significant effects on the central nervous system.

As with all studies conducted in model organisms, there are some specific contextual factors that make comparison to humans difficult. While zebrafish are used to model complex brain disorders, including anxiety, limitations exist because we must infer pain, discomfort or other behaviors through observation [21, 22, 81, 82]. Another unique aspect of zebrafish exposure is ammonia in water. To address this potentially toxic metabolite, we regularly measured ammonia and found no effect of nitrate or nitrite treatment on water ammonia levels. Due to the large number of animals needed to conduct the study, we were limited in the number of doses we could test and thus focused on a nitrate dose and exposure duration associated with improvements in exercise performance [27]. Another limitation of our study is that it was conducted in healthy adult zebrafish thus it is not clear what effect nitrate or nitrite treatment would have in the presence of conditions like neurodegenerative disorders. More and larger studies are needed to delineate the potential benefits and risks associated with nitrate and/or nitrite treatment on CBF, mood, and cognitive function, particularly in populations of people with differing ages and underlying health status. Importantly, a study in humans is underway to look at the effect of increasing doses of nitrate on cognition-related outcomes [83]. We also cannot differentiate between the direct effects of nitrate or nitrite in the fish, or indirect effects that could be generated by increased NO availability. Nevertheless, we show that nitrate and nitrite treatment in a zebrafish model did not adversely affect multiple parameters of health but was associated with mild anxiogenic-like behavior, changes in brain metabolome, and an initial decrease in executive function or associative learning.

## Supporting information

**S1 Fig. Experiment design summary.**
(TIFF)

**S1 Table. Metabolites annotated using our in-house library.**
(XLSX)

**S2 Table. Significantly changed metabolites, between all treatment groups.**
(XLSX)

## Acknowledgments

We thank Lindsey St. Mary, Eric Johnson, Carrie L. Barton, Sabrina Edwards, and Kimberly Hayward (Sinnhuber Aquatic Research Laboratory), Claudia S. Maier (Department of Chemistry and OSU Mass Spectrometry Center), and Jeffrey Morrè (Operational Manager, Oregon State University Mass Spectrometry Center) for technical assistance and advice.

## Author Contributions

**Conceptualization:** Manuel García-Jaramillo, Laura M. Beaver, Lisa Truong, Elizabeth R. Axton, Rosa M. Keller, Robyn L. Tanguay, Jan F. Stevens, Norman G. Hord.

**Data curation:** Manuel García-Jaramillo, Laura M. Beaver, Lisa Truong, Rosa M. Keller.

**Formal analysis:** Manuel García-Jaramillo, Laura M. Beaver, Lisa Truong, Rosa M. Keller, Kathy R. Magnusson.

**Funding acquisition:** Robyn L. Tanguay, Jan F. Stevens, Norman G. Hord.

**Investigation:** Manuel García-Jaramillo, Laura M. Beaver, Lisa Truong, Elizabeth R. Axton, Rosa M. Keller, Mary C. Prater.

**Methodology:** Manuel García-Jaramillo, Laura M. Beaver, Lisa Truong, Elizabeth R. Axton, Rosa M. Keller, Robyn L. Tanguay, Jan F. Stevens, Norman G. Hord.

**Project administration:** Laura M. Beaver, Lisa Truong, Elizabeth R. Axton, Rosa M. Keller.

**Resources:** Lisa Truong.

**Software:** Manuel García-Jaramillo, Laura M. Beaver, Lisa Truong.

**Supervision:** Jan F. Stevens, Norman G. Hord.

**Validation:** Manuel García-Jaramillo, Laura M. Beaver, Lisa Truong, Kathy R. Magnusson, Jan F. Stevens.

**Visualization:** Manuel García-Jaramillo, Laura M. Beaver, Lisa Truong, Rosa M. Keller, Norman G. Hord.

**Writing – original draft:** Manuel García-Jaramillo, Laura M. Beaver, Lisa Truong.

**Writing – review & editing:** Manuel García-Jaramillo, Laura M. Beaver, Lisa Truong, Elizabeth R. Axton, Rosa M. Keller, Mary C. Prater, Kathy R. Magnusson, Robyn L. Tanguay, Jan F. Stevens, Norman G. Hord.

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
