## [Decision Letter · Decision Letter 0]

20 Oct 2020

PONE-D-20-27164

Nitrate and nitrite exposure increases anxiety-like behavior and alters brain metabolomic profile in zebrafish

PLOS ONE

Dear Dr. García - Jaramillo,

Thank you for submitting your manuscript to PLOS ONE. As you will see, the reviewers were generally supportive of the publication of this work. However, both reviewers raised several major concerns that preclude publication of the paper in its present form. In particular, one of the reviewers recommends re-considering whether to include all of the behavioural measures, which they found somewhat confusing and difficult to understand in the context of this study, in the final manuscript. Therefore, after careful consideration, we feel that it has merit but does not fully meet PLOS ONE’s publication criteria as it currently stands. We invite you to submit asubstantially revised version of the manuscript that addresses the points raised during the review process.

We look forward to receiving your revised manuscript.

Kind regards,

Matthew Parker

Academic Editor

PLOS ONE

Journal Requirements:

2. As part of your revisions, please discuss all methods undertaken to ameliorate potential pain and distress, e.g. proper monitoring of behavior, humane endpoints, etc.

Reviewers' comments:

Reviewer's Responses to Questions

**Comments to the Author**

1. Is the manuscript technically sound, and do the data support the conclusions?

Reviewer #1: Yes

Reviewer #2: No

2. Has the statistical analysis been performed appropriately and rigorously? 

Reviewer #1: Yes

Reviewer #2: Yes

3. Have the authors made all data underlying the findings in their manuscript fully available?

Reviewer #1: Yes

Reviewer #2: Yes

4. Is the manuscript presented in an intelligible fashion and written in standard English?

Reviewer #1: Yes

Reviewer #2: Yes

5. Review Comments to the Author

Reviewer #1: The manuscript entitles “Nitrate and nitrite exposure increases anxiety-like behavior and alters brain metabolomic profile in zebrafish” by García-Jaramillo is well-written and brings new information about how nitrate and nitrite affect learning and memory. Although the manuscript is interesting and new, there are few comments that need to be answered.

1. The authors used fish from a range of 9 to 16 months old. Zebrafish reach adult age around 3-4 months old, here the authors used fish from 9 to 16 months. It is known that aging affects cognitive performance and it is slightly worrying that the authors used a range of 7 months difference to this experiment. My question is why did the authors use fish from 9 to 16 months old? Were the ages equally distributed between groups?

2. Line 171. “Fish were fed a standard lab diet (Gemma Micro. Skretting, Westbrook, ME) at a volume of ~3% 172 body weight/day.” Did the authors feed the fish only once a day? If yes, why? Fish feeding regimen is really important and can modulate metabolism and behavior (doi.org/10.7717/peerj.5343).

3. Lines 173 – 175. Please explain the methods used to collect the fish body for further hormone analysis.

4. Lines 189 – 191 - The behavior was recorded between 14-17 days but it is not clear when each test was performed. I would recommend that the authors create an experiment design figure to make it clear for the readers.

5. In the figure legends, the N varies from 42 to 84 which is a big difference. Why for some groups is there almost double the sample size? Please describe in detail the sample size calculation and any criteria for exclusion. This is an important part of the research design and should always be decided prior. Also, the information about removal of outliers is essential for research transparency as well as how many fish were excluded.

6. The authors need to improve the result section, focusing on three points. More descriptive analysis is necessary, 1) the F and P values must be added in the result section since only on line 321 it was added; 2) The degrees of freedom and denominator values must be added; 3) Add the descriptive information about the negative results as well.

7. Lines 418 – 419 – The authors must discuss their data with care. Although they found that they disproved the hypothesis that nitrate and nitrite improve cognitive performance, they used healthy adult fish and no model of cognitive deficits was tested in this manuscript.

8. Although the discussion about the possible hypothesis of migraines being associated with behavioral response is interesting, I wonder if there is any behavioral characterization of migraine models? e.g. how nitroglycerin induced migraine’s affect fish behavior? It feels like there is a missing discussion about the specific link between behavior and migraine’s, since the authors only discussed about the physiological effects, but mention it as a possible factor that could affect behavior.

9. Please improve the term “mild anxiety-like behavior” because it does not indicate whether it is an anxiolytic or anxiogenic effect, this term is used in the abstract, discussion and conclusion.

Reviewer #2: This paper does have something to offer the field. The behavioral data are largely problematic, and while I have some suggestions for improving some of the measures, I think the simplest solution would be to exclude many of them from the paper. This will then require some reformulation of the discussion and conclusions, but ultimately I think that there are enough results of substance to merit publication. Therefore, my recommendation is to reconsider publication after revision.

General question – Upon reading the abstract and opening of the paper, my first question was one that was not addressed until the very end of the paper – that of known adverse effects of nitrate pollution, and concerns over ammonia exposure. The hypothesis that nitrate and nitrite exposure should enhance performance in fish was therefore surprising. While a full discussion of this issue could reasonably wait until the end, curious readers might appreciate an early acknowledgement of this apparent discrepancy when the question is introduced.

General Conclusions – The experiment primarily presents null results, used as evidence that nitrate and nitrite do not substantially affect health, learning or behavior. This can be useful, but extra precautions must be used (in terms of appropriate controls and experimental rigor) to ensure that the null results are not simply a failure of the experimental design to detect an effect. Without an especially high methodological standard, null results are not very informative. My suggestions below are geared at reducing the number of uninterpretable results stemming from less rigorous procedures, to allow greater focus on the positive and potentially meaningful results.

Habituation - The statistics seem to show that there is no difference in habituation, but simply that there is a slight reduction in startle response in the nitrate-treated fish.

*Suggested resolution – modify the wording of the discussion and interpretation to make clear that a difference in startle reactivity was observed, but that no difference was detected in habituation.

Avoidance - The avoidance procedure and results are not reported in enough detail to determine the validity of the results. For example, it doesn’t appear as though there were any controls in place to rule out differences in locomotor activity or sensitization induced by the shock, or sensory/attentional deficits (e.g. reduced attention to the blue light). If this is the case, while there might be mild behavioral differences between treatment groups, those differences cannot be attributed to learning or cognitive differences (they could simply be reactivity to or perception of the stimuli, such as the startle effects observed in the habituation data). I have looked up the reference (34) which explains the shuttlebox technique in more detail, but this also does not indicate whether an appropriate control was used (it seems not).

*Suggested Resolution – include results from appropriate controls, or remove this treatment completely from the manuscript. The results cannot be interpreted without the control groups, and may only provide a premise for future studies to use similar procedures.

Social/Predator - I am concerned about the validity of the social/predator test. If the animals were behaving as predicted, the ‘No stimulus’ condition should provide a neutral baseline from which the ‘social’ condition should show an increase, and the ‘predator’ condition should show a decrease in proximity. Instead, both stimulus conditions reflect a mild reduction in proximity compared to no stimulus, making it unclear how the fish perceive the video stimuli. The control treatment group did not show any significant response to the videos, making it especially difficult to interpret the changes observed in the treatment groups (which were strongest in the no-stimulus condition). In the absence of interpretable preferences by the control condition, the changes observed in the treatment condition are not very informative.

*Suggested Resolution – I would remove these data from the manuscript, again to avoid establishing a precedent of a procedure that doesn’t really measure what it is intended to measure. I think they could be left in with plenty of caveats and discussion of the results (since there was a clear effect of treatment), but it seems that this wouldn’t strengthen the paper, and instead simply distract from the more interesting effects.

Tank Location – This does seem to be the one relatively straightforward behavioral effect of treatment. However, it would be necessary to statistically compare ‘Time in Zone – bottom’ across the three treatment groups to determine whether the treatment does lead to significantly more time near the bottom. Because analysis revealed an interaction, a post-hoc test that confirms this difference would be warranted – I simply don’t see the results of such a test reported.

*Suggested Resolution – include post-hoc analysis for duration in the bottom zone across treatment groups. If this difference is not significant, adjust language in the discussion to (further) soften conclusions related to anxiety.

I am surprised that the authors tested catecholamines, but not cortisol, which would be a logical assay for stress or anxiety-related behaviors. I don’t see the inclusion of cortisol levels as necessary, but it would have enhanced the discussion and might be considered for next time.

Minor suggestions:

In many places when reporting the statistics, the wording seems stilted – usually around phrases involving statistical significance. Check these and see if they can be re-phrased.

Examples: Line 345: “a statistically significant more time close to the monitor”

line 338 "associated with a significant higher percentage of fish"

The discussion begins with a statement that “we disproved the hypothesis that nitrate, and nitrite treatment would improve indicators of learning and cognitive performance in a zebrafish model” This is too strong a statement for results that were largely null – instead, the study simply did not find evidence supporting the hypothesis (it similarly did not find evidence against the hypothesis). If the changes suggested above are made, all reference to learning/cognition should be carefully revised.

The final result, indicating that neither nitrate nor nitrite was taken into the brain in any great quantity, deserves more attention. For example, it seems odd to discuss at length the role of NO in learning and memory when the effects of treatment are likely to be through indirect mechanisms (as acknowledged near the end). Similarly, the section on migraines is a very interesting speculation, but again rather lengthy given the absence of direct evidence from this study (it might be a worthwhile future direction). The apparent effects of treatment on purine metabolism, annotated fatty acids, GABA and Glutamine are fairly straightforward, and the bulk of the discussion should focus on possible pathways and mechanisms for the effects that were observed.

6. PLOS authors have the option to publish the peer review history of their article (what does this mean?). If published, this will include your full peer review and any attached files.

Reviewer #1: **Yes: **Barbara D. Fontana

Reviewer #2: **Yes: **Rachel Blaser

---

## [Author Response · Author response to Decision Letter 0]

3 Dec 2020

Response to reviewers’ comments

The comments of both reviewers on our manuscript were very helpful to us in clarifying our paper. We have carefully read each of these comments and will respond to them below. We would like to thank them for their positive comments. Furthermore, the suggestions made by the reviewers have been extremely helpful in the revision of this manuscript. To address these suggestions and provide further clarity in the text we have revised the manuscript to incorporate the reviewers’ suggestions. The changed text is marked by highlights and is found throughout the paper. Below you will find the detailed reviewers’ comments and our responses.

Reviewer #1: The manuscript entitles “Nitrate and nitrite exposure increases anxiety-like behavior and alters brain metabolomic profile in zebrafish” by García-Jaramillo is well-written and brings new information about how nitrate and nitrite affect learning and memory. Although the manuscript is interesting and new, there are few comments that need to be answered.

Response: Thank you for your helpful comments regarding our manuscript. We appreciate your time in careful review, and the positive encouragement. We have responded to each comment in the sections below and made the appropriate changes in our resubmission.

1. The authors used fish from a range of 9 to 16 months old. Zebrafish reach adult age around 3-4 months old, here the authors used fish from 9 to 16 months. It is known that aging affects cognitive performance and it is slightly worrying that the authors used a range of 7 months difference to this experiment. My question is why did the authors use fish from 9 to 16 months old? Were the ages equally distributed between groups?

Response: We agree that age can affect cognitive performance. When our experiments were performed, we took fish of the same age and equally distributed them between treatment groups in order to alleviate the potential impact of age on our results. We have added additional text to the methods section to clarify this. The reason for the age variation is that we ran the experiment in multiple cohorts using available fish. All fish used to conduct this study were healthy adults that still had reproductive capacity. 

2. Line 171. “Fish were fed a standard lab diet (Gemma Micro. Skretting, Westbrook, ME) at a volume of ~3% 172 body weight/day.” Did the authors feed the fish only once a day? If yes, why? Fish feeding regimen is really important and can modulate metabolism and behavior (doi.org/10.7717/peerj.5343).

Response: We agree this is important. We fed them twice a day and fish were never more than 4h from a feeding when behavior experiments were conducted. We added further details about it to the methods section.

3. Lines 173 – 175. Please explain the methods used to collect the fish body for further hormone analysis.

Response: Thank you for pointing this out. We added text and a reference to the methods section in regard to this question. Briefly, for hormone analysis the animals were euthanized using rapid cooling directly prior to collecting the samples according to Wallace et al (2018).

CK Wallace, LA Bright, JO Marx, RP Andersen, MC Mullins, AJ Carty. Effectiveness of Rapid Cooling as a Method of Euthanasia for Young Zebrafish ( Danio rerio). J Am Assoc Lab Anim Sci. 2018 Jan 1;57(1):58-63. PMID: 29402353 PMCID: PMC5875099

4. Lines 189 – 191 - The behavior was recorded between 14-17 days but it is not clear when each test was performed. I would recommend that the authors create an experiment design figure to make it clear for the readers.

Response: Thank you for your comment. We have added additional text in the methods section to clarify when each test was performed. We have also included an experiment design summary in supplemental information (S1 Fig).

5. In the figure legends, the N varies from 42 to 84 which is a big difference. Why for some groups is there almost double the sample size? Please describe in detail the sample size calculation and any criteria for exclusion. This is an important part of the research design and should always be decided prior. Also, the information about removal of outliers is essential for research transparency as well as how many fish were excluded.

Response: Thank you for comment. We agree with the reviewer that it was not very clear in our previous version. With n=84 we meant that we had measured that parameter twice and thus had double the number of observations. Since Reviewer#2 also expressed some concerns about this, we have removed the additional data from the manuscript. To further answer the reviewers concern about outliers, we confirm that no outliers were excluded in our analysis. 

6. The authors need to improve the result section, focusing on three points. More descriptive analysis is necessary, 1) the F and P values must be added in the result section since only on line 321 it was added; 2) The degrees of freedom and denominator values must be added; 3) Add the descriptive information about the negative results as well.

Response: Thank you for your comment. Following reviewer recommendation, we have excluded some of the behavioral measures from our final version, in order to better focus the results section and make it easier to understand. Following PLOS ONE’s guidelines, we have included F and P values within the figures; the degrees of freedom and denominator values are not requested within this Journal guidelines. 

7. Lines 418 – 419 – The authors must discuss their data with care. Although they found that they disproved the hypothesis that nitrate and nitrite improve cognitive performance, they used healthy adult fish and no model of cognitive deficits was tested in this manuscript.

Response: Thank you for your comment. We agree with the reviewer and have changed this sentence to soften it and added the point about a limitation as it relates to only testing healthy fish in the discussion.

8. Although the discussion about the possible hypothesis of migraines being associated with behavioral response is interesting, I wonder if there is any behavioral characterization of migraine models? e.g. how nitroglycerin induced migraine’s affect fish behavior? It feels like there is a missing discussion about the specific link between behavior and migraine’s, since the authors only discussed about the physiological effects, but mention it as a possible factor that could affect behavior.

Response: We appreciate the reviewer’s question and did evaluate the available research on this topic. The group that has used nitroglycerin as a model for migraine published two papers on its effects in common carp (Cyprinus carpio L.). While they do show effects of treatment on blood brain barrier permeability and brain edema, they unfortunately, did not test for changes in behavior. We have edited this section of the discussion following reviewer’s recommendation.

9. Please improve the term “mild anxiety-like behavior” because it does not indicate whether it is an anxiolytic or anxiogenic effect, this term is used in the abstract, discussion and conclusion.

Response: Thank you for your comment. We have made the change per the reviewer’s suggestion from “mild anxiety-like behavior” to “mild anxiogenic-like behavior”.

Reviewer #2: This paper does have something to offer the field. The behavioral data are largely problematic, and while I have some suggestions for improving some of the measures, I think the simplest solution would be to exclude many of them from the paper. This will then require some reformulation of the discussion and conclusions, but ultimately I think that there are enough results of substance to merit publication. Therefore, my recommendation is to reconsider publication after revision.

Response: Thank you for your time in critical review and the positive recommendation. We have responded to each comment in the sections below and made the appropriate changes in our resubmission. We think our manuscript has been improved because of your careful review and suggestions.

General question – Upon reading the abstract and opening of the paper, my first question was one that was not addressed until the very end of the paper – that of known adverse effects of nitrate pollution, and concerns over ammonia exposure. The hypothesis that nitrate and nitrite exposure should enhance performance in fish was therefore surprising. While a full discussion of this issue could reasonably wait until the end, curious readers might appreciate an early acknowledgement of this apparent discrepancy when the question is introduced.

Response: Thank you this suggestion. We added a sentence to the introduction to address this concern.

General Conclusions – The experiment primarily presents null results, used as evidence that nitrate and nitrite do not substantially affect health, learning or behavior. This can be useful, but extra precautions must be used (in terms of appropriate controls and experimental rigor) to ensure that the null results are not simply a failure of the experimental design to detect an effect. Without an especially high methodological standard, null results are not very informative. My suggestions below are geared at reducing the number of uninterpretable results stemming from less rigorous procedures, to allow greater focus on the positive and potentially meaningful results.

Habituation - The statistics seem to show that there is no difference in habituation, but simply that there is a slight reduction in startle response in the nitrate-treated fish.

*Suggested resolution – modify the wording of the discussion and interpretation to make clear that a difference in startle reactivity was observed, but that no difference was detected in habituation.

Response: Thank you for your comment. We have made this change in the results and discussion sections.

Avoidance - The avoidance procedure and results are not reported in enough detail to determine the validity of the results. For example, it doesn’t appear as though there were any controls in place to rule out differences in locomotor activity or sensitization induced by the shock, or sensory/attentional deficits (e.g. reduced attention to the blue light). If this is the case, while there might be mild behavioral differences between treatment groups, those differences cannot be attributed to learning or cognitive differences (they could simply be reactivity to or perception of the stimuli, such as the startle effects observed in the habituation data). I have looked up the reference (34) which explains the shuttlebox technique in more detail, but this also does not indicate whether an appropriate control was used (it seems not).

*Suggested Resolution – include results from appropriate controls, or remove this treatment completely from the manuscript. The results cannot be interpreted without the control groups, and may only provide a premise for future studies to use similar procedures.

Response: Thank you for your comment. We apologize for the lack of details on the shuttlebox. The shuttlebox technique has an appropriate control where the stimuli (e.g. light) is not at a fixed location in the box. Depending on where the fish is in the box, the light opposite is turned on. This serves as a way to rule out reactivity or habituation per a fish. Additionally, there is 30 trials per fish, which reduces the probability of the behavioral response being attributed to cognitive differences, and more likely learning behavior.

Social/Predator - I am concerned about the validity of the social/predator test. If the animals were behaving as predicted, the ‘No stimulus’ condition should provide a neutral baseline from which the ‘social’ condition should show an increase, and the ‘predator’ condition should show a decrease in proximity. Instead, both stimulus conditions reflect a mild reduction in proximity compared to no stimulus, making it unclear how the fish perceive the video stimuli. The control treatment group did not show any significant response to the videos, making it especially difficult to interpret the changes observed in the treatment groups (which were strongest in the no-stimulus condition). In the absence of interpretable preferences by the control condition, the changes observed in the treatment condition are not very informative.

*Suggested Resolution – I would remove these data from the manuscript, again to avoid establishing a precedent of a procedure that doesn’t really measure what it is intended to measure. I think they could be left in with plenty of caveats and discussion of the results (since there was a clear effect of treatment), but it seems that this wouldn’t strengthen the paper, and instead simply distract from the more interesting effects.

Response: Thank you for your comment. Yes, we agree with the Reviewer and removed this data per your suggestion.

Tank Location – This does seem to be the one relatively straightforward behavioral effect of treatment. However, it would be necessary to statistically compare ‘Time in Zone – bottom’ across the three treatment groups to determine whether the treatment does lead to significantly more time near the bottom. Because analysis revealed an interaction, a post-hoc test that confirms this difference would be warranted – I simply don’t see the results of such a test reported.

*Suggested Resolution – include post-hoc analysis for duration in the bottom zone across treatment groups. If this difference is not significant, adjust language in the discussion to (further) soften conclusions related to anxiety.

Response: Thank you for your recommendation. We have completed post-hoc analysis for duration in the bottom zone across treatment groups and found no significant differences. For this reason, we have softened the language to point this out.

I am surprised that the authors tested catecholamines, but not cortisol, which would be a logical assay for stress or anxiety-related behaviors. I don’t see the inclusion of cortisol levels as necessary, but it would have enhanced the discussion and might be considered for next time.

Response: Thank you for your recommendation. We agree with the reviewer and will test cortisol in any future studies. Our analysis of the brain metabolomics did not allow us to identify cortisol; we will utilize a targeted approach in future studies to assist with identifying this important hormone in this model system. 

Minor suggestions:

In many places when reporting the statistics, the wording seems stilted – usually around phrases involving statistical significance. Check these and see if they can be re-phrased.

Examples: Line 345: “a statistically significant more time close to the monitor”

line 338 "associated with a significant higher percentage of fish"

Response: Thank you for your comment. We have removed the language associated with line 345 and revised the paragraph that was stilted and contained line 338. 

The discussion begins with a statement that “we disproved the hypothesis that nitrate, and nitrite treatment would improve indicators of learning and cognitive performance in a zebrafish model” This is too strong a statement for results that were largely null – instead, the study simply did not find evidence supporting the hypothesis (it similarly did not find evidence against the hypothesis). If the changes suggested above are made, all reference to learning/cognition should be carefully revised.

Response: Thank you for your comment. We made the change to the hypothesis statement per the reviewer’s suggestion. 

The final result, indicating that neither nitrate nor nitrite was taken into the brain in any great quantity, deserves more attention. For example, it seems odd to discuss at length the role of NO in learning and memory when the effects of treatment are likely to be through indirect mechanisms (as acknowledged near the end). Similarly, the section on migraines is a very interesting speculation, but again rather lengthy given the absence of direct evidence from this study (it might be a worthwhile future direction). The apparent effects of treatment on purine metabolism, annotated fatty acids, GABA and Glutamine are fairly straightforward, and the bulk of the discussion should focus on possible pathways and mechanisms for the effects that were observed.

Response: Thank you for your comment. We have restructured the discussion to emphasize the indirect nature of the associations with nitrate and nitrite exposure on behavior. We explore the potential effects on neuronal function and behavior due to changes in these metabolites as suggested. The discussion now reflects a broader discussion of potential associations between nitrate and nitrite exposure based on the observed behavioral changes and alterations in brain metabolomic profile. 

6. PLOS authors have the option to publish the peer review history of their article (what does this mean?). If published, this will include your full peer review and any attached files.

Do you want your identity to be public for this peer review? For information about this choice, including consent withdrawal, please see our Privacy Policy.

Reviewer #1: Yes: Barbara D. Fontana

Reviewer #2: Yes: Rachel Blaser

---

## [Decision Letter · Decision Letter 1]

14 Dec 2020

Nitrate and nitrite exposure leads to mild anxiogenic-like behavior and alters brain metabolomic profile in zebrafish

PONE-D-20-27164R1

Dear Dr. García - Jaramillo,

We’re pleased to inform you that your manuscript has been judged scientifically suitable for publication and will be formally accepted for publication once it meets all outstanding technical requirements.

Kind regards,

Matthew Parker

Academic Editor

PLOS ONE

Additional Editor Comments (optional):

Reviewers' comments:

Reviewer's Responses to Questions

**Comments to the Author**

1. If the authors have adequately addressed your comments raised in a previous round of review and you feel that this manuscript is now acceptable for publication, you may indicate that here to bypass the “Comments to the Author” section, enter your conflict of interest statement in the “Confidential to Editor” section, and submit your "Accept" recommendation.

Reviewer #1: All comments have been addressed

2. Is the manuscript technically sound, and do the data support the conclusions?

Reviewer #1: Yes

3. Has the statistical analysis been performed appropriately and rigorously? 

Reviewer #1: Yes

4. Have the authors made all data underlying the findings in their manuscript fully available?

Reviewer #1: Yes

5. Is the manuscript presented in an intelligible fashion and written in standard English?

Reviewer #1: Yes

6. Review Comments to the Author

Reviewer #1: The authors have significantly improved the manuscript and therefore I recommend the article to be accepted.

7. PLOS authors have the option to publish the peer review history of their article (what does this mean?). If published, this will include your full peer review and any attached files.

Reviewer #1: No

---

## [Editor Report · Acceptance letter]

18 Dec 2020

PONE-D-20-27164R1 

Nitrate and nitrite exposure leads to mild anxiogenic-like behavior and alters brain metabolomic profile in zebrafish 

Dear Dr. García-Jaramillo:

I'm pleased to inform you that your manuscript has been deemed suitable for publication in PLOS ONE. Congratulations! Your manuscript is now with our production department. 

Kind regards, 

on behalf of

Dr. Matthew Parker 

Academic Editor

PLOS ONE